# Underground Pipeline Identification into a Non-Destructive Case Study Based on Ground-Penetrating Radar Imaging

**Nicoleta Iftimie ***, **Adriana Savin** **, Rozina Steigmann** **and Gabriel Silviu Dobrescu**

National Institute of Research and Development for Technical Physics, 47 D. Mangeron Blvd,
700050 Iasi, Romania; asavin@phys-iasi.ro (A.S.); steigmann@phys-iasi.ro (R.S.); gdobrescu@phys-iasi.ro (G.S.D.)
* Correspondence: niftimie@phys-iasi.ro; Tel.: +40-232-430-680

**Abstract:** Ground-penetrating radar (GPR) has become one of the key technologies in subsurface sensing and, in general, in nondestructive testing (NDT), since it is able to detect both metallic and nonmetallic targets. GPR has proven its ability to work in electromagnetic frequency range for subsoil investigations, and it is a risk-reduction strategy for surveying underground various targets and their identification and detection. This paper presents the results of a case study which exceeds the laboratory level being realized in the field in a real case where the scanning conditions are much more difficult using GPR signals for detecting and assessing underground drainage metallic pipes which cross an area with large buildings parallel to the riverbed. The two urban drainage pipes are detected based on GPR imaging. This provides an approximation of their location and depth which are convenient to find from the reconstructed profiles of both simulated and practical GPR signals. The processing of data recorded with GPR tools requires appropriate software for this type of measurement to detect between different reflections at multiple interfaces located at different depths below the surface. In addition to the radargrams recorded and processed with the software corresponding to a GPR device, the paper contains significant results obtained using techniques and algorithms of the processing and post-processing of the signals (background removal and migration) that gave us the opportunity to estimate the location, depth, and profile of pipes, placed into a concrete duct bank, under a structure with different layers, including pavement, with good accuracy.

**Keywords:** ground-penetrating radar; nondestructive testing; pipelines detection; modeling; signal processing

## 1. Introduction

One of the most effective and powerful nondestructive testing (NDT) employed in road surveys nowadays is the ground-penetrating radar (GPR), due to its high flexibility of usage and reliability of results. A reliable risk-reduction strategy to pipe examination is the key for ensuring the sustainable development and improvement of the life time of urban water supply and drainage system. The drainage pipes are critical endowment for a smart city as a precursor for reaching a sustainable development and having a limited life time. The pipes age with functioning time being buried deep underground and can lead to significant safety hazards as water dissipation and soil contamination. These possible disadvantages affect day-to-day use and the long-lasting life of urban pipes. Among NDT inspection techniques is laser scanning, which is geospatial method that can detect only spatial distribution visible in the acquisition, while ultrasound elastic waves, a geophysical method, is capable to detect only quantitative data about failures and cannot achieve more characteristics of pipes [1].

GPR is a geophysical technique based on very short electromagnetic pulses (1–20 ns) propagation within radiofrequency band, typically between 10 MHz to few GHz, to map profile and underground features [2,3]. GPR has numerous characteristics, providing a high resolution, strong anti-interference ability, and high efficiency. It is also a nondestructive

technique; consequently, GPR has been extensively used in many fields, such as geological exploration, water conservancy engineering, and urban construction [4,5]. The properties depending on frequency (dielectric permittivity ($\varepsilon$), electrical conductivity ($\sigma$) and magnetic permeability ($\mu$)) play a significant role in the electromagnetic energy dissipation in a medium containing complex discontinuities [3–5].

Until now, GPR surveys are widely employed as a noninvasive detection tool to detect unknown targets in deep underground and is useful in the localization of electromagnetic (EM) discontinuities in the subsurface with high resolution [6,7].

Interesting applications fields of GPR are measurements for underground targets location (ex. cables, landmine and UXO, drainage pipes) as in archaeology [8], civil engineering [9,10], military applications [11–13], etc. The transducers realize the coupling of energy within the near field by evanescent and propagating waves [14].

The GPR equipment record the time between the sending of the impulse and its receiving after the scattering the emitted waves which undergo several propagation processes. The main registered is reflected by interfaces, while some is also scattered and returns to the surface, and the signals are presented in B-scan radargram [11,15,16].

The main interest is to detect as many characteristics as is possible to image a buried object [14], and to extract its clear image from ground discontinuities [17,18]. With GPR technology underground, drainage pipelines can be detected and it has been used also in civil engineering to evaluate major structural damage, such as holes and cavities in roads, plates, and bridge decks [19,20]. Estimating the location of damaged pipes is significant for service performance and sustainable management.

A new infrastructure has been developed in Iasi, Romania, on the terraces of the riverbed that crosses the city and includes an upper terrace (underground various targets and drainage pipes detection in this proposed), a lower terrace (detection of pipes for sewer leaks [21]), and a medium terrace (investigations related to spill basins and civil protection dam [22]. The land area along to the riverbed is particularly important, both for the design of civil constructions and for infrastructure. The problem becomes even more important given the action of seismic movements, as these lands have different behavior from the usual situation. Furthermore, for the case of concrete/asphalted tracks, the fatigue life of the urban pipes located under the road/pedestrian area represents a problem of high importance, because they are close to urban heating system and utility water ducts.

This paper presents a case study employing GPR signals for detecting and assessing underground drainage metallic pipes which cross an area with large buildings parallel to the riverbed. The research exceeds the case study at the laboratory level and was realized in the field in a real case where the scanning conditions are much more difficult due to the fact that the area is not perfectly straight and the weather conditions are not always favorable. Reflections of electromagnetic waves occur and are created in places or layers of the ground where a variation of electrical or magnetic properties occurs. In areas with different water content and with buried pipes and tunnels, there is a variation in the speed of propagation of radar waves, and strong reflections occur. With the help of techniques and algorithms for processing and post-processing the appropriate signals (migration and background removal), the noises are eliminated and the shape and depth of the investigated objects are rendered to a significant extent. The work was performed in the frame of a large project (starting from 2015 to present) with the aim of transforming the area around river bank into an ecological agreement park. The project targeted all areas that were at risk by restoring the infrastructure with different types of buried pipes for water transport or sewage leakage, overflow basins, and the civil protection dam [21,22].

This paper presents the results recorded with a GPR tool to detect drainage pipes with unknown approximate position buried under a bike track on the river bank, and a succession of digital signal processing and post-processing methods applied both to A-scan and B-scan that provide an easy way to read and interpret the results. The proposed methods and the algorithms are considered for the recognition and detection of a concrete duct bank containing two drainage pipes for hot water transportation (turn-return). The

situation has been simulated using the finite-difference time-domain (FDTD) [2] software in order to interpret the signals recorded by receiving GPR antenna, bowtie type, working at 400 MHz [23–25].

## 2. Materials and Methods

### 2.1. Generic GPR with Bowtie Antenna and Propagation Waves

The electromagnetic methods as GPR are non-invasive where individual measurements are quasi real-time, and due to the relatively low frequencies used, have the advantage of penetrating electromagnetic waves at great depths in the ground and obtaining scattering information from buried bodies at great depths. As it is known, the resolution is defined as the capacity of the measurement system to discriminate individual elements embedded in a different medium [11]. The penetration depth decreases with the increase in the conductivity of the medium and, for higher frequency, high resolution and lower penetration depths are obtained. Figure 1 presents a standard GPR principle consist of a transmitting (Tx) and a receiving (Rx) antennas placed in a shielded case which is displaced over the surface to be scanned [26,27]. The GPR emits EM waves that penetrate in the ground in the form of an ellipse (*inset bottom* Figure 1).

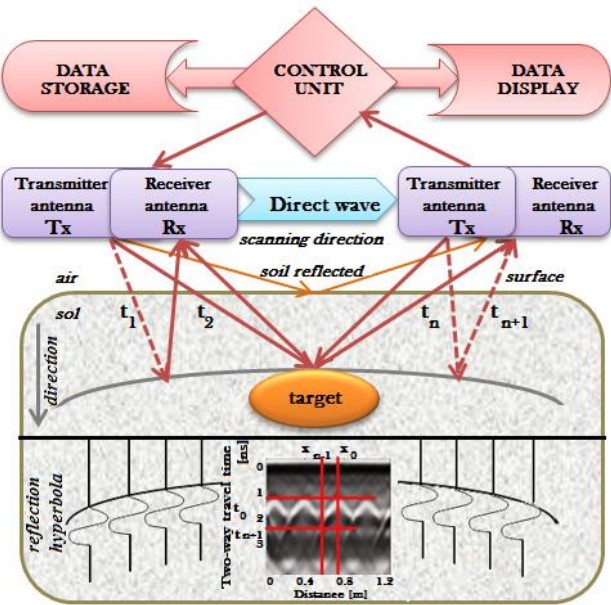

**Figure 1.** Process of a generic GPR system (*inset bottom*-schematic diagram of GPR reflection hyperbola generation and real signal GPR B-scan).

After being recorded and processed by the control unit embedded in a GPR system, the reflected waves were mixed into a reflection signal A-scan, measuring the interval between emission and reception of the signal delivered by the reception transducer (Figure 1). It was predicted that the time of flight $t$ of the GPR signal will be double—forward and backward to the buried target at $z$ depth, and the remaining constant during the survey [2,28,29] will increase as the distance between transmitter and receiver increases ($x$) (*inset bottom* Figure 1). Because the area scanned with GPR had mostly multi-layer targets, the speed had to be calibrated according to layer $n$ [2,30]

The hyperbola in the radar signature when the radar is moving along X-axis (propagating medium is considered homogeneous) [31] is given by

$$R = \sqrt{z_0^2 + (X - x_0)^2} \tag{1}$$

where $(x_0, z_0)$ is a perfect point scattered in the 2D plane, $X$ is the synthetic aperture vector, and $R$ is the path length vector (from antenna to scattered). As mentioned in [21],

simulation techniques that comprise single frequency models, time domain models, ray tracing, integral techniques, and discrete element methods may be useful in foreseen the results to be obtained from in-field measurements. The FDTD technique is one of the simulation methods that is most suitable with GPR surveys [32,33].

The propagating waves (homogeneous waves—the wavenumber is real [34]) in the near field of the transducer determined the coupling of the energy into the ground. The nonmagnetic soil ($\mu_s = \mu_0$) in which the pipes are buried has electrical properties, relative permeability ($\varepsilon_{rs}$), and conductivity ($\sigma_s$), and the field induced by antenna has the features of the rectangular coil [7]. For a region free of sources, the Helmholtz equation is useful according to [35].

The field generated by the emission coil feed by a current with frequency [36] can be expressed using dyadic Green's function [37] and integral method

$$\overline{E}_0(\bar{r}) = j\omega\mu_2 \int\limits_{Vsource} \overset{\leftrightarrow}{G}_{12}(\bar{r},\bar{r}\prime)\overline{J}(\bar{r}\prime)d\bar{r}\prime, \tag{2}$$

where $\mu_2$ is magnetic permeability of the medium 2 (soil as Figure 1 depicts), $\overset{\leftrightarrow}{G}_{12}$ is component of dyadic Green's function matrix. The electric conductivities considered are $\sigma_f$ for the target and respective $\sigma_2$ for the stratified soil [38] and the total electric field become

$$\overline{E}_2(\bar{r}) + j\omega\mu_2\sigma_2 \int\limits_{Vbody} \overset{\leftrightarrow}{G}_{22}(\bar{r},\bar{r}\prime)\overline{E}_2(\bar{r}\prime)\left[\frac{\sigma_f(\bar{r}\prime)}{\sigma_2} - 1\right]d\bar{r}\prime = \overline{E}_0(\bar{r}), \tag{3}$$

and perturbation field in air in the presence of conductive target is according with [39]

$$\overline{E}_1(\bar{r}) = j\omega\mu_2\sigma_2 \int\limits_{Vbody} \overset{\leftrightarrow}{G}_{21}(\bar{r},\bar{r}\prime)\overline{E}_2(\bar{r}\prime)\left[\frac{\sigma_f(\bar{r}\prime)}{\sigma_2} - 1\right]d\bar{r}\prime \tag{4}$$

where $\overset{\leftrightarrow}{G}_{12}$, $\overset{\leftrightarrow}{G}_{22}$ are components of dyadic Green's function matrix [38].

### 2.2. Geophysical Surveys in Pavement Assessment and Drainage Water Pipes Detection

The GPR equipment used is Utility Scan Standard System (Geophysical Survey Systems, Inc. GSSI, Nashua, NH, USA) (Figure 2a) [40], which had a 400-MHz antenna, allowing a penetration depth until 4.5 m depending on the moisture of soil. The front wheel of utility scanner had an encoder, allowing a displacement precision determination of $\pm1$ mm, and the scan intervals assured by the GSSI System software was 100 scans/m. The sampling rate was 0.04 ns, with the quantization of the signal being made on 16 bits [40]. The equipment was set up to record A-scan at each 10 cm, and the time window for which the signals were obtained is 32 ns. The control unit contained a function that allowed the testing of the terrain dielectric by recording a data set and then performing its migration. By knowing the soil permittivity, the penetration depth could also be implicitly known. For example, the user manual showed that a profile of at least 3 m in length is collected over well-known objects and that they have to go through those objects at a right angle [40]. By using the up and down arrows, we could adapt the dielectric value for that hyperbolas profiles crack-up to points.

The dielectric constant of the soil in the survey area was established at $\varepsilon_r = 12$, in the basis of previous investigations [21,22] knowing the exactly type of soil. Due to the weather conditions (rainy), the value of the dielectric constant was taken from a table from the user manual of GSSI equipment and tested with TEST_DIEL function [41].

A region of [2000 × 600] cm from the Bahlui river bank was surveyed (Figure 2b) in the immediate vicinity of the riverbed. The urban drainage pipes for hot water was taken into study (Figure 3). The pipes had an unknown approximate position, being buried under a bike track. We assumed that they were buried parallel with the riverbed.

Due to its orientation and practical survey procedures, the scanning was performed in 3 parallel traces with a 2000-cm length, separated between them with 100 cm, as seen in Figure 3a. The scanning of directions orthogonal on the direction of riverbank was facilitated. Figure 3c presents a photo of the testing zone when the new pipes were brought to be replaced.

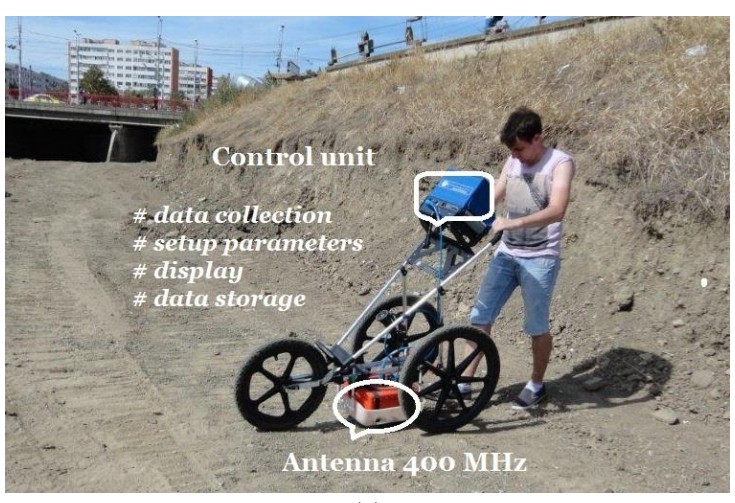

(**a**)

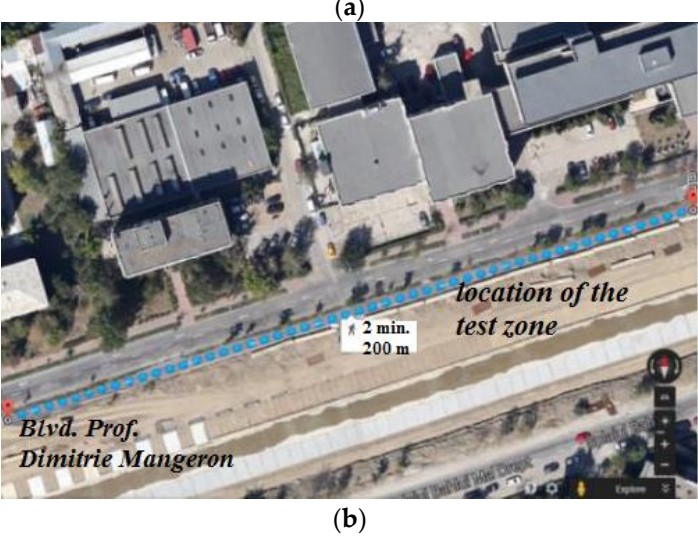

(**b**)

**Figure 2.** Experimental set-up: (**a**) GPR survey system with a 400-MHz antenna; (**b**) test area of the underground urban drainage pipes.

The urban drainage pipes had an approximate 70-cm diameter and, for hot water transportation, had 15-cm wall thickness of insulation (nonwoven glass fiber). In order to simplify the data presentation, a zone of [600 × 500] cm was selected and the scanning was effectuated in 6 transversal traces with 600-cm length, separated between them with 100 cm (the traces were effectuated in both directions, see Figure 3b).

*2.3. Signal Processing: A-Scan and B-Scan*

The mean value to A-scan data set were crucially assured to be close to zero, so that the amplitude probability distribution from A-scan data set was symmetric to the mean value [2].

$$A\prime_n = A_n - \frac{1}{N}\sum_{n=1}^{N} A_n, \tag{5}$$

where $A_n$ are values of raw data set, $A'_n$ are values of processed data set, $n$ is the data set number, and $N$ represents total number of data sets.

The filtering operation is given by

$$A\prime_n = A_n + \frac{A_n - A\prime_{n-1}}{K},\tag{6}$$

where $A'_n$ is averaged value, $A_n$ is the current value.

The $K$ factor will be chosen to take values from $n$ to $N$ or a fixed value, and will contribute to the average value. Averaging has no effect on discontinuities.

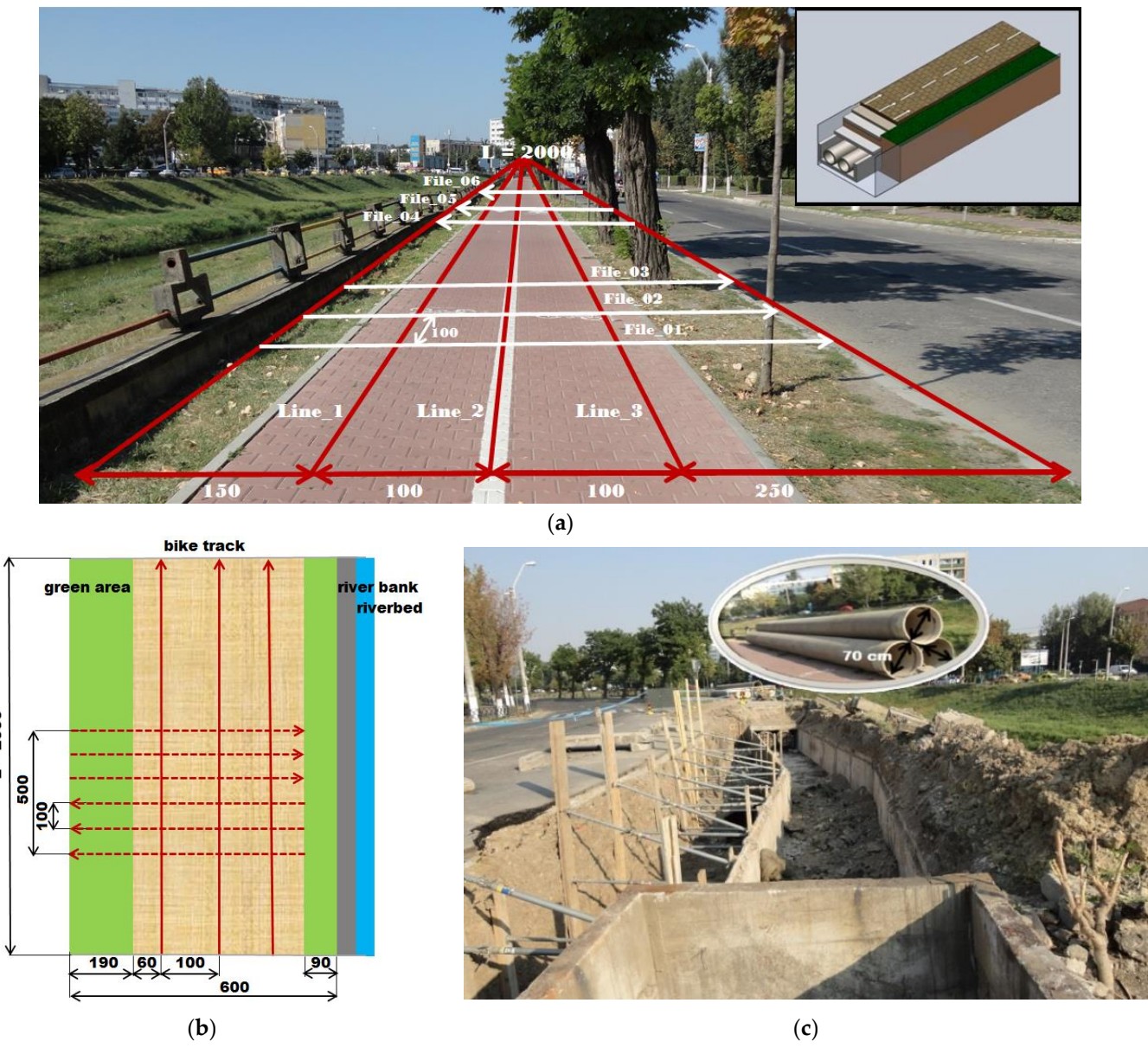

**Figure 3.** Surveyed zone: (**a**) the investigated region on both directions on drainage pipes (inset—configuration of the surveyed zone); (**b**) scanning scheme on both directions; (**c**) the region where the two urban drainage pipes should be repaired or replaced (inset-pipes to be replaced); (**c**) picture was taken several days after GPR surveying); (all dimensions are in cm).

Thus, a window of $L$ pixels was defined, and the mean of the pixels in it from all the pixels in this window was subtracted. The window advanced and the procedure was repeated until the entire image is solved, as

$$g(x,y) = f(x,y) - \frac{1}{L} \sum_{i=-L/2}^{i=L/2} f(x+i,y), \tag{7}$$

where $g$ is filtered image, $f$ is raw data, and $L$ is the window size.

Using nonlinear optimization of the decomposition technique, [41] improves GPR imaging by simultaneously determining spatial variations in size and delaying soil reflection.

Considering $A_i(x)$, the spatially soil reflection amplitude, and $B_i(x)$, the time delay of the soil reflection apex over segment $i^{th}$, then $A_i(x)$ and $B_i(x)$ could be approximated as a sum

$$A_i(x) = \sum_{n=0}^{4} a_{i_n} T_n(x); \quad B_i(x) = \sum_{n=0}^{4} b_{i_n} T_n(x) \ , \tag{8}$$

where $T_n(x)$ are the Chebyshev polynomials established by the recursive relation.

$$T_{n+1}(x) = 2T_n(x) - T_{n-1}(x), \quad n > 1 \ , \tag{9}$$

with $T_0(x) = 1$ and $T_1(x) = x$.

### 3. Results

*3.1. Application of GPR Data Raw and FDTD Simulations in the Detection and Replace of Water Pipes*

As we showed in previous researches [11,21,22], a GPR device with the corresponding control unit can record a continuous image of the subsurface, which indicates the presence, depth, and the layout of soil features required in classification, characterization, and surveying of soil as well as detection and identification of various buried targets.

GPR waves are modified by the subsurface layer and the recorded radar data sets a contrast in electrical and magnetic properties; those changes can then be detected, represented, and characterized. A GPR data set recording delivers high-resolution information that is able to use at the interpretation and the extrapolation of information obtained with algorithms and pre-processing techniques. Figure 4 presents the scan on longitudinal directions of a zone where two urban drainage pipes with known diameter placed in a concrete duct are buried. Only one reflection peak can be seen, given by direct-coupling by the A-scan raw, which takes place when the antenna is lightly displaced from the soil [42–44]. In this case, the direct waveform transmitting and receiving antenna in connected with the surface to produce a mixed waveform. A signal (A-scan) with a deep reflection to record above concrete described in Figure 4a. It can also be seen that, in the case of real data set, the signal is very noisy, including clutters [21]. B-scans were obtained from 55 raw A-scan types, using the specific signal processing, similar to those of ultrasound examinations, as presented in Figures 4 and 5.

Figure 4 shows the results which were divided by subheadings to assure a succinct and accurate evaluation of the data set recording, their interpretation, as well as the preliminary conclusions that could be drawn. At the distance of 44 m from the starting point (Figure 4b), a signal with the form of a distorted hyperbola and a peak pointing upwards is observed at the depth of 20.2 ns. This is indicating the fact that the drainage pipes with the axes parallel with the scanning direction change their orientation.

Figure 5a,b shows the presence of the two drainage pipes for hot water transportation, with an 85-cm diameter (70-cm diameter of the pipe and 15-cm protective layer), both buried in concrete duct ($\varepsilon_s = 8$) [41], the top of pipes being at a 20.2-ns depth. It first uses the GPRMax software to produce GPR synthetic datasets through FTDT simulations [31]. The simulated data was processed with a code in Matlab 2020b (MathWorks, Inc., Natick, MA, USA). In Matlab, we used functions for removing noise by adaptive filtering, for example, "wiener2", which filters the grayscale image using a pixel-wise adaptive low-pass

Wiener filter, and a pixelwise adaptive Wiener method based on statistics estimated from a local neighborhood of each pixel. The waves penetrating the ground are propagated along the scanning line and produce EM pulses at manually chosen intervals and detect buried targets. The next reflected EM pulse can be incorporated within a radargram B-scan to produce an underground 2D image. Figure 6 presents the simulation of the experimental set-up and surveying conditions using GPRMax 2D. The layout of the survey is previewed with GPRMaxGV—Gnuplot viewer—a free plot script auto formatter developed by Goran Bekic [45]. The pipelines are difficult to identify in noisy profiles, which means that their GPR patterns are sensitive to noises. Introducing the raw radargrams as the test set into the simulation model, a positive section, corresponds favorable with the object locations. A reliable detection depth of GPR is determined by the central frequency of the electromagnetic wave and the attribute of the subsurface formation, while detection resolution is limited by the wave frequency and signal bandwidth [7].

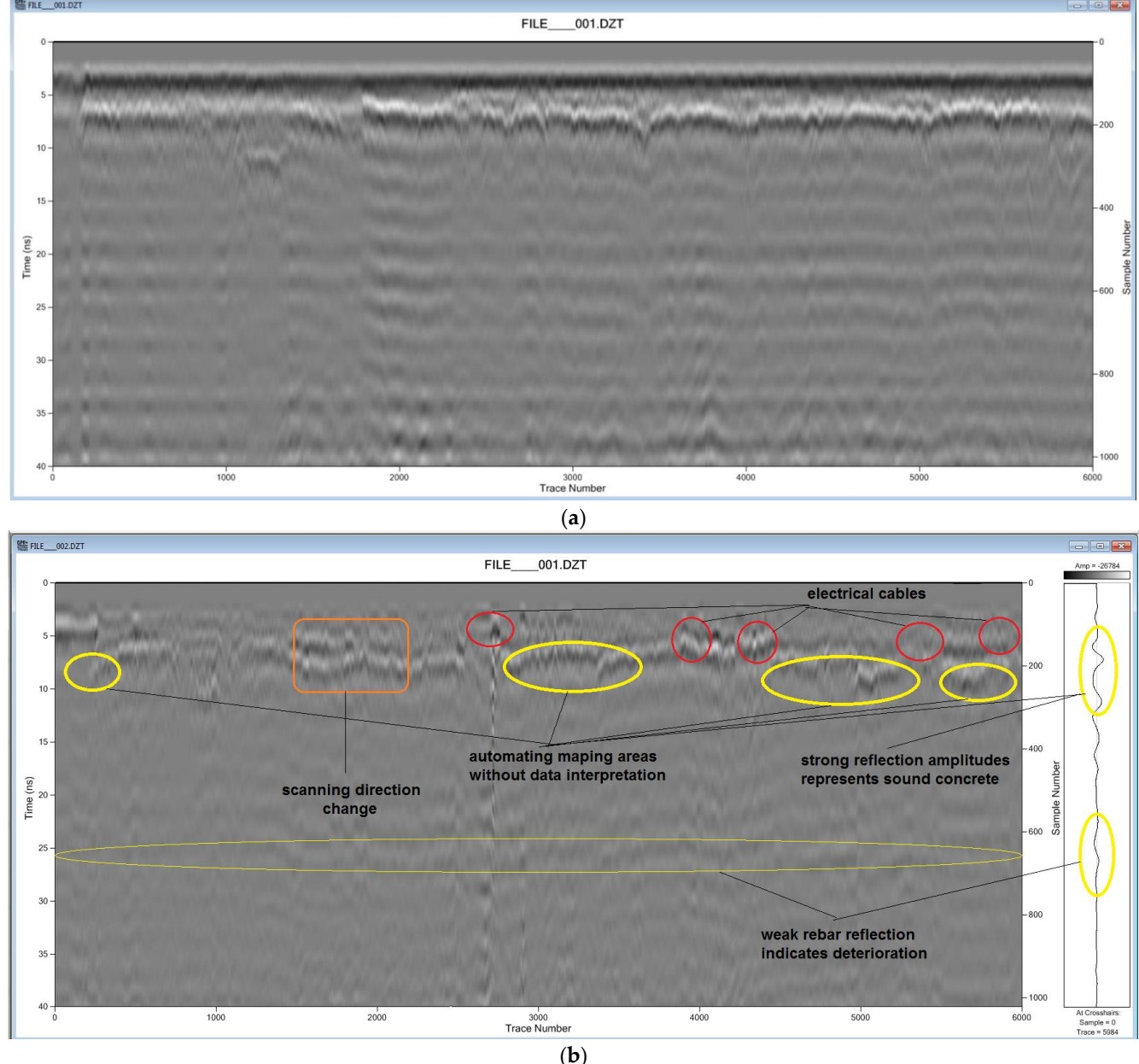

**Figure 4.** (**a**) GPR raw data containing a profile with the air wave, (**b**) GPR radargrams of surveyed zone—parallel traces and partition for data interpretation.

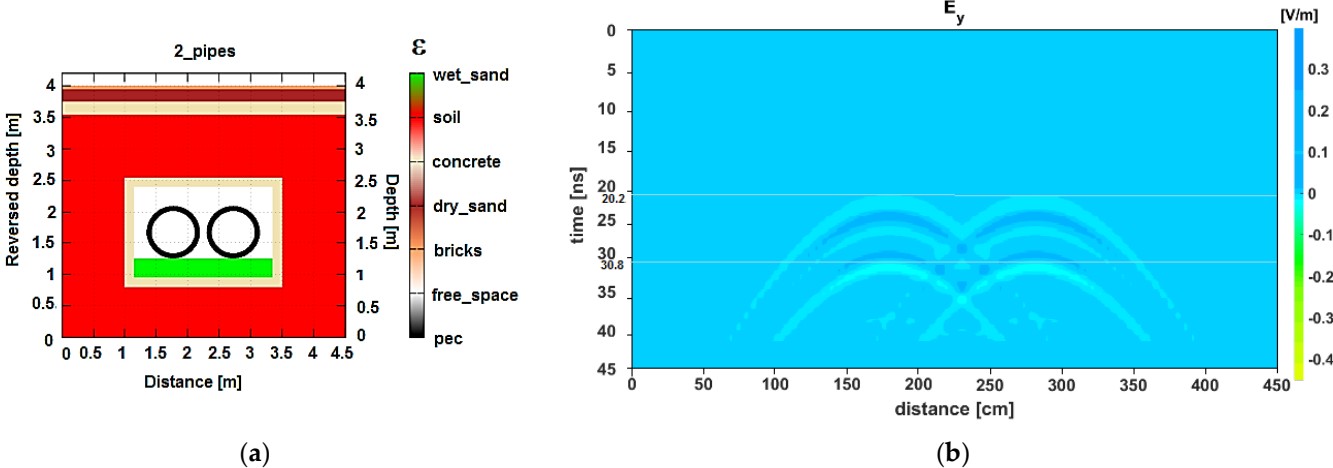

**Figure 5.** Raw data imaging at scan of zone transversal on drainage pipes (see Figure 3a): (**a**) File_001_002_003 forward scan, (**b**) File_004_005_006 backward scan.

**Figure 6.** Simulation using GPRMax 2D (**a**) the simulated geometry and visualization using GPRMaxGV Gnuplot viewer; (**b**) Signal processing of simulated data.

### 3.2. A-Scan and B-Scan Results

An original A-scan is presented in Figure 7a, while Figure 7b presents the processed A-scan where the mean value was zero and the noise was decreased according to Equation (6). The value of K was selected as 1.02 according with [21] on the basis of probability of detection principle and of the characteristics of reception, where *K* is a measure of interpretation to eliminate the conflict between two sequential acquisition of data and to keep at least 2% error, according to [6]. Considering a set of five samples containing a B-scan, a series of techniques of signal processing can be taken into account. Usually, the clutters hamper the imaging of GPR data. In order to make the evaluation of GPR radargrams as accurate and correct as possible, extraction of no-longer-desired signals as retransmission of wave from Tx to Tr or reflections in the air–soil interface must be effectuated. This is named background removal; good results can be obtained using a subtract mean trace procedure [46].

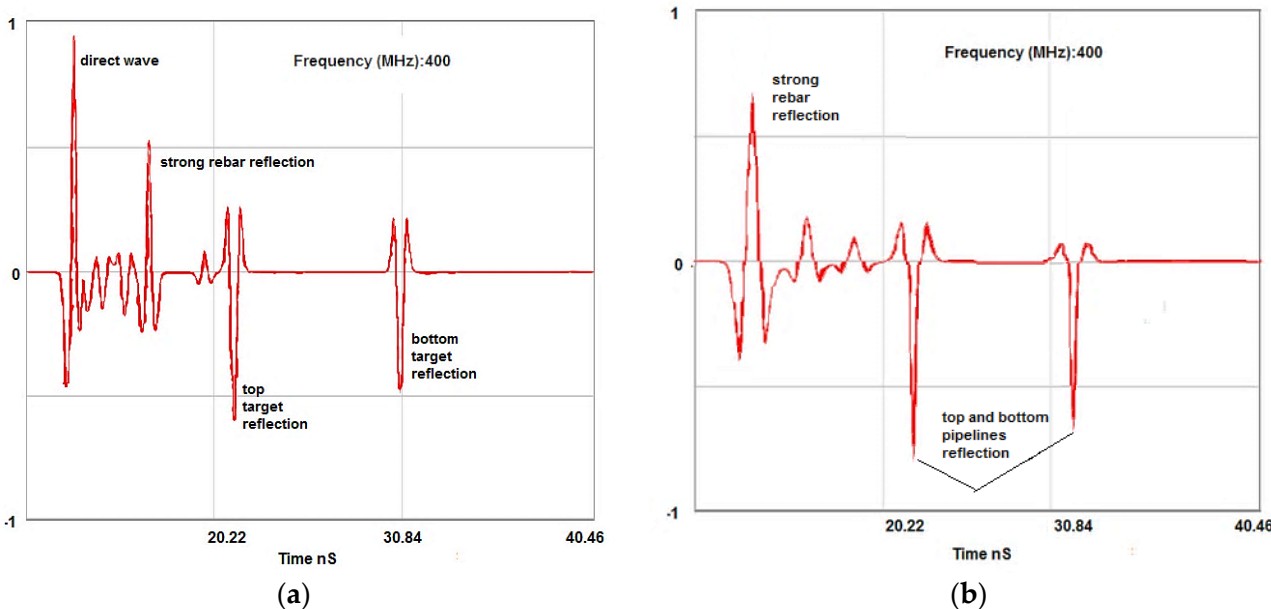

**Figure 7.** A-scan after filtering of raw experimental data delivered by GPR system: (**a**) original GPR record (raw data with direct wave); (**b**) after filtered signal (show both rebar and the pipelines reflections).

After the weighting coefficients ($\overline{a_i}$, $\overline{b_i}$) was calculated, the amplitude *Ai(x)* and the delay *Bi(x)* of segment $i^{th}$ were determined with Equation (5). Parameter calculation was carried out with a nonlinear least square error minimization function, in the Matlab 2020b Optimization Toolbox. If the optimization does not converge for a data set, we must use a recursive approach. Imaging procedures can be used to concentrate the energy existing in a point target's hyperbolic arc which returns to a unique point [41]. This technique, named migration, allows the establishment of the depths of underground objects. Figure 8 present B-scan results processing of one file over the scanned area. For a B-scan original image from the inspected area, the algorithm presented above [21] is applied. Figure 8a shows the results after background removal with L = 20 pixels (length of sliding Hanning window). Figure 8b presents the result after application of migration technique using the Kirchhoff method with 0.15 m/ns. Top and bottom reflections of concrete duct bank, in which the two drainage pipes are located, could be remarked at approximately 20.2 ns, 30.8 ns from the data set. The parallel line with scanning direction at 30.8 ns describes the reflections on the remnant water from the pipes and can be seen on the processed image from the experimental data. The rest of parallel lines were due to numerous reflections from the soil scanned zone and on the interface concrete duct bank of the urban drainage pipes. In the same area, we can notice the signals whose shape may indicate the presence of buried

electrical cables (for example, at 98.6 m, 128.5 m, 153.3 m, 169 m, etc.). On the basis of raster scan results, the GPR-slice function from matGPR software (open-source software) was used to obtain a 3-D volume of GPR data and to enable its visualization in the form of opaque or translucent slices [47,48] In GPR operation, the high resolution in depth is obtained by utilizing a transmitted signal of wideband. The high resolution was obtained by coherently processing scattered electromagnetic waves which were measured while the device scanned along a line of terrain. The resolution after the postprocessing depended on the focusing capacity of the migration algorithm.

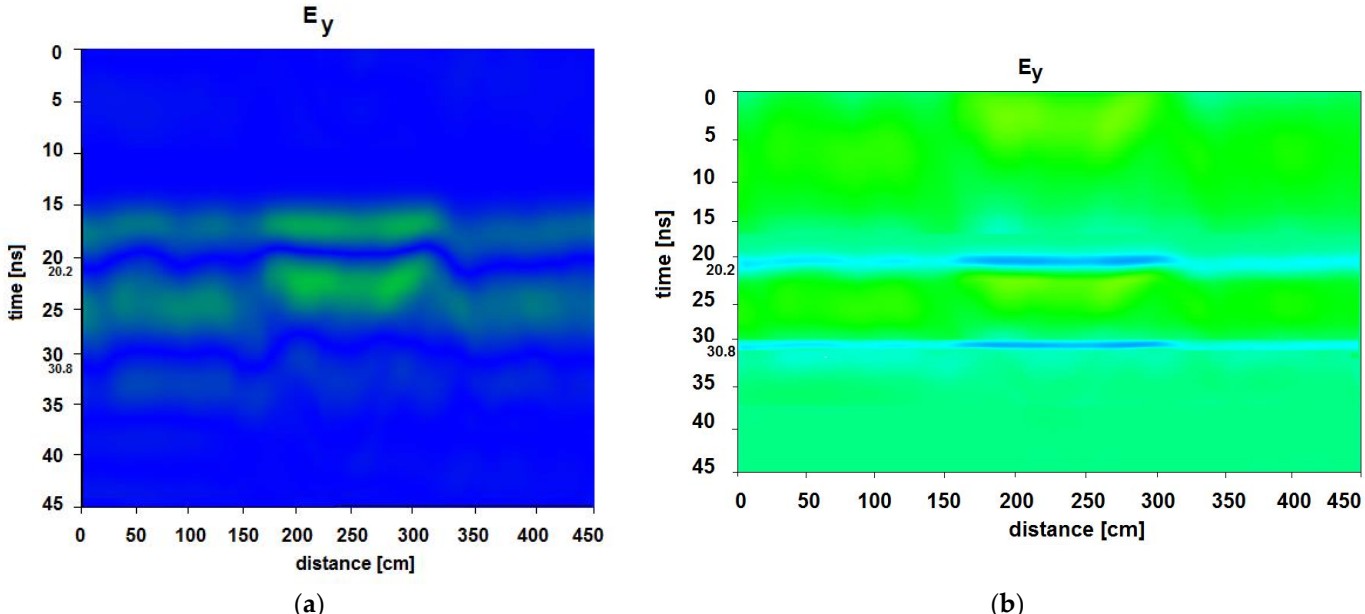

**Figure 8.** B-scan raw data processed after using both techniques: (**a**) ground removal; (**b**) migration.

It can be observed that, with each processing step, an improvement of resolution was obtained, helping to emphasize the delimitation of bottom and upper level of the pipes.

B-scan radargrams were concatenated as a result of the parallel line positioned at 20.2 ns, allowing the representation of a 3D dataset under the form of planes (isometric surfaces) [49–52] with equal signal, generating and displaying an orthographical designing of the subsurface equal to 55% of the maximum signal included in the 3D data set volume (Figure 9).

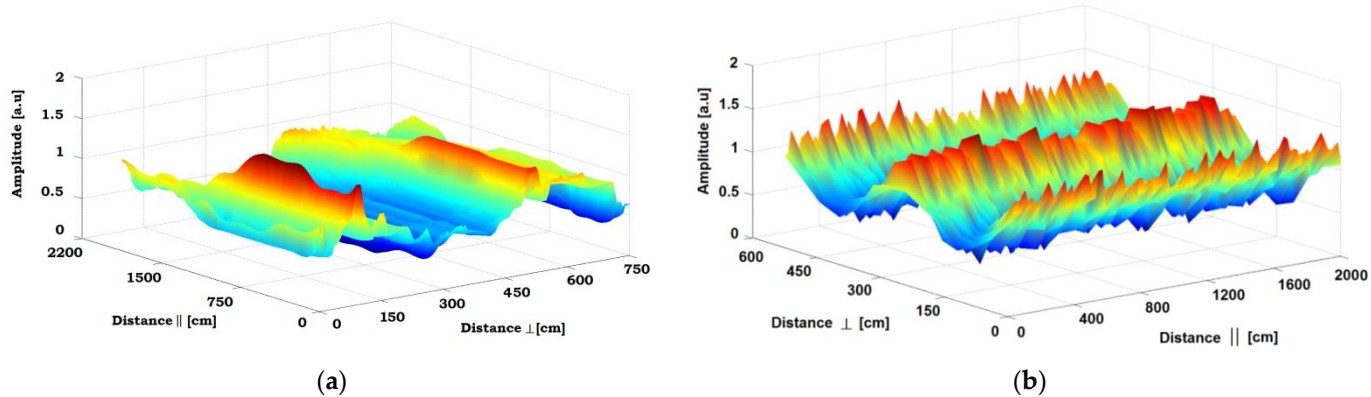

**Figure 9.** Drainage water pipes reconstruction: (**a**) before and (**b**) after processing.

## 4. Discussions

Most researches [12,13,15] on object detection only determined whether the objects were detectable and where the objects were. Other researches have determined the sizes of rebars and small-scale voids [3,5,8,42] inside concrete, but investigations inside the complicated underground sections become difficult. Giannikis et al. [33] attempted to estimate the buried object size (buried ammunitions, including landmine), but they only utilized numerically produced data. Elsewhere, Grimberg et al. [11] determined the location and size subsurface buried ammunitions, including antitank mines. Jin et al. [20] developed a machine learning framework based on wavelet scattering networks to analyze GPR data for subsurface pipeline identification with smaller diameters. In our previous works, we also detected pipes for sewer leaks [21] and investigated spill basins and civil protection dams [22]. This paper presents the results of a case study employing GPR signals for detecting and assessing underground drainage metallic pipes which cross an area with large buildings parallel to the riverbed. The research exceeds the case study at the laboratory level, which is accomplished in the field in a real case where the scanning conditions are far more difficult. After surveying the zones, the GPR radargram recording signals of drainage pipes with an unknown approximate position were buried under a bike track on the river bank. It used a succession of digital signal processing and post-processing methods applied both to A-scans and B-scans which provide an easy way to read and interpret the results. The simulations were made with FDTD software to aid interpretation of the signals recorded by receiving a GPR antenna. The profile of the drainage pipes was obtained and their diameters could then be estimated. Numerical modelling and signal post-processing algorithms of GPR can lead to a better understanding of the operating principle of the radar detection tools [47,53]. GprMax allows the simulation of real cases of GPR in order to gain an idea of what is expected during surveys and to improve complex signal processing and interpretation skills until receiving the real data [9,10]. The propagation waves theory is applied to the GPR detection of drainage pipes, and the signal processing technique is used for A-scans and B-scans of recorded dataset [21,22]. In our study, simulations were made with FDTD software and survey data were recorded by GPR system, working at 400 MHz where the location, depth, and profile of pipes could be determined.

## 5. Conclusions

Using GPR raw data and techniques and algorithms of processing and post-processing of the signals (background removal and migration), the obtained results provided the opportunity to estimate the location, depth, and profile of pipes, placed into a concrete replaceable duct bank. High-frequency electromagnetic waves can recognize underground objects in depth due to sharp energy attenuation. Accepting the noise sensitivity of pipelines and the failure in recognizing the deep pipeline, the techniques and algorithms used in these study case presents promising applicability in both simulated and practical GPR signals. The ground removal and migration methods were introduced and evaluated for comparison study of GPR B-scan image processing and gave us the opportunity to estimate location, depth, and profile of pipes. Processing of the acquired data was carried out with Matlab software, which is in the endowment of our laboratory and not in a standard application of a commercial software suite available and used by geophysicists, because of the technical features of our survey. Based on Matlab software, the B-scan data were converted into images, thus highlighting the depth and position of the buried pipes and the shape that can be evaluated with good accuracy. Future research will focus on the knowledge of the physical, mechanical, and chemical properties of the lands in the investigates area as well as the analysis of environmental pollution risks by testing areas, including runways, platforms, perimeter road, and road handling, which represent decisive factors both in the design of civil engineering and for infrastructure works.

**Author Contributions:** Conceptualization, N.I., A.S. and R.S.; methodology, N.I., A.S. and R.S.; software, R.S. and G.S.D.; validation, N.I. and A.S.; investigation, N.I. and G.S.D.; writing—original draft preparation, N.I. and A.S.; writing—review and editing, N.I., A.S., R.S. and G.S.D.; supervision, N.I. and A.S.; All authors have equal contributions. All authors have read and agreed to the published version of the manuscript.

**Funding:** This paper is partially supported by Protocol IUCN 365/11.05.2021, Theme 04-4-1142-2021/2025 Project 30; Romanian Ministry of Research, Innovation and Digitization CCCDI - UEFIS-CDI, project number PN-III-P2-2.1-PED-2019-2148, within PNCDI III, contract no. 568PED/2020 - Innovative models of violins acoustically and aesthetically comparable to heritage violins – MINOVIS and Nucleus Program PN 19 26 01 02. And The APC was funded by Protocol IUCN 365/11.05.2021, Theme 04-4-1142-2021/2025 Project 30; Romanian Ministry of Research, Innovation and Digitization CCCDI - UEFISCDI, project number PN-III-P2-2.1-PED-2019-2148, within PNCDI III, contract no. 568PED/2020 - Innovative models of violins acoustically and aesthetically comparable to heritage violins – MINOVIS. and Nucleus Program PN 19 26 01 02.

**Institutional Review Board Statement:** Not applicable.

**Informed Consent Statement:** Not applicable.

**Data Availability Statement:** Not applicable.

**Conflicts of Interest:** The authors declare no conflict of interest.

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
