# Peer review of "Underground Pipeline Identification into a Non-Destructive Case Study Based on Ground-Penetrating Radar Imaging"

_remotesensing, doi:10.3390/rs13173494_

Round 1

Reviewer 1 Report

This paper describes well known problem of towns' underground utility finding. Sometime GPR gives good results, sometimes it doesn't have success. This depends on concrete environment. The paper is interesting but it is not too innovative.

Reviewer 2 Report

Review of the manuscript remotesensing-1340235 “Electromagnetic technique applied in underground pipeline identification based based on Ground Penetrating Radar imaging” by Nicoleta Iftimie, Adriana Savin, Rozina Steigmann and Gabriel Silviu Dobrescu.

This manuscript is a resubmission of a previous version that I had the opportunity to proceed to the revision. I would like to thank for the changes that were made and that are presented in this version under review. In the previous review, I suggested a set of recommendations, related to the organization of the manuscript structure and the improvement of some contents. The weaknesses presented prevent us from understanding what the original contribution of this publication is.

This review allowed us to prove that some of the suggestions were accepted, which is an improvement to the proposed article. However, the current version still has problems that must be resolved. The main problem that I identify is the lack of clarity about the novelty that is intended to be communicated with this publication, which was not clarified in the new version under review. Below are my comments on the review.

Regarding the title of the article, I have a question: what is the electromagnetic technique presented? Reading the article reveals that the authors use GPR to study the subsurface where pipes are buried. For this reason, the title should mention that this is a study of a place with a technique that already exists (GPR), that is, it is a case study.

In the abstract, line 20-21, "the paper contains news information obtained using techniques and algorithms that gave us the opportunity to estimate". What is new technique and new algorithm? The reading does not reveal the conception of a new technique or a new algorithm.

In keywords, line 29, "electromagnetic induction" is not appropriate.

In line 44-45, "Among NDT inspection techniques such as laser scanning which is a geospatial method is able to detect only spatial distribution", should clarify that the laser scanner only allows detecting the spatial distribution that is visible in the acquisition.

In line 52, "between 10 MHz to few GHz", you should clarify the amount of GHz. This information is known, it can be mentioned as it sets the lower value.

In line 59, "containing dielectric discontinuities", you should remove the word dielectric.

In line 70, "after the scattering", it should be clarified that the emitted waves undergo several propagation processes. The main registered is reflected by interfaces, some is also scattered and returns to the surface.

In line 72, "clutter", this term is not appropriate. The subsurface structure can be homogeneous.

On line 77, "our city", you must clarify which is your city (and country?).

In line 87, "This paper presents a case study using GPR signals for detection and assessing of underground drainage metallic pipes which crosses an area with large buildings parallel to the riverbed.", this is the true objective of the study. The entire article must be restructured considering this crucial information.

In line 104, "The paper proposes to present the results", you should delete the word proposes, because the article presents the results. Here again the authors mention that the article presents the results of the GPR acquisition.

In line 119, "bowtie antenna", I don't understand the meaning of bowtie.

The whole section 2.1 has too much formalism that I don't think contributes positively to conveying the message of the publication. The information described is not new. Thinking about presenting the method in a summarized way and only with the strictly necessary formalism to convey the intended idea.

In line 136, "in ground an ellipse", this part of the sentence is confusing.

In line 185, section “2.3” is not well written, the numbering does not follow the previous value. Check and rectify.

Figure 4 has some misunderstanding. Part (a) if it is raw data must contain a profile with the air wave. Part (b) turned out better than in the previous version, but the gain can be increased to see the details you want to show. Pay attention to the size of the labels, so that they are legible on an A4 printed sheet. The horizontal axis has number of dashes when it can have the distance as in the previous version.

In line 245, "using some specific signal processing", you should state what they are.

In Figure 5, you must increase the gain so that you can see the important details. I cannot see anything. Care must be taken when presenting graphical outputs. Pay attention to the size of the labels.

In line 254, "at the depth of 20.2 ns", I cannot see it. Check or improve graphics output.

In Figure 6, the air wave is masking the reflections corresponding to the pipes of the synthetic model. If you remove it and adjust the gain it will show better. Note that in the simulation there is no noise because it considered a homogeneous medium. Pay attention to the statements that start on line 273. There is no noise to hinder the visualization of everything in the synthetic profile. The gain does not allow you to see it, but it can improve with adjustment.

Section 3.2 must move to the chapter where it presents the method. Chapter 3 is for results. Pay attention to formalism. You should mention the formalism that is necessary to explain the ideas you want to convey.

In Figure 7 shows the effect of filtering with data that have the air wave. Filtering should be applied after removing the air wave.

In line 320, "a improved technique is proposed", which one? I cannot understand the proposal of a new technique in this article.

In Figure 8, the improvement introduced by migration is not observed. About the migration operation, which was applied? Was it the one described by reference [62] as it is written in line 320? Is a new migration operation different from the ones that already exist? If it is not new, it is necessary to remove the formalism between lines 320 and 341.

In Figure 9, I suggested that part (a) be an image before processing. The image to be considered should be a 3D image as in part (b) but with data before processing, to be able to compare the improvement of the results. Otherwise, it is not possible to compare with different graphic outputs. Some scheme can be added to help interpret.

In line 387, "in order to interpret", the simulation program simulates data to aid interpretation. Improve sentence.

In line 408, "proposed method", what is the proposed method? There is a substantive issue in the article that was identified in the previous review that has not been resolved.

In line 417-419, "The processing of the acquired data has been carried out in Matlab, not in a standard application of a commercial software suite available for geophysicists, because of the technical features of our survey, investigators are physicists.", don't understand, physicists can't use commercial programs? This sentence must be improved to make sense.

About the conclusions, these do not show what is new in the article. The authors says that a new technique was presented but he doesn't understand what it is. This means that the article must be reviewed to resolve this serious issue.

An individualized discussion is still lacking. Chapter 3 should be for results only. Furthermore, there is no real discussion of the results to compare with other approaches that predate the article. This flaw contributes to the thesis that this article does not reveal any novelty in terms of the method conceived. The article can be reformulated to present a case study, which is pertinent for a publication, and thus there is no problem with the lack of novelty in the proposed method, as it presents a result and not a method.

To conclude, figure captions must have a brief description of what is observed in the images.

Reviewer 3 Report

The paper has been appropriately revised and I think it should be accepted

Round 2

Reviewer 2 Report

Second review of the manuscript remotesensing-1340235 “Electromagnetic technique applied in underground pipeline identification based based on Ground Penetrating Radar imaging” by Nicoleta Iftimie, Adriana Savin, Rozina Steigmann and Gabriel Silviu Dobrescu.

This manuscript is a resubmission of a previous version that I had the opportunity to proceed to the revision. I would like to thank for the changes that were made and that are presented in this version under review.

The authors responded to all questions and suggestions. The article is now almost ready to be published, and it is only necessary to make a small effort to resolve some weaknesses that I identified:

  1. Line 212: EDTD maybe is a mistake with FTDT.
  2. Line 289: Automatic reference with a error message.
  3. Figure 4: The (b) part still have problems; the gain is very low, so the reflections can't suggest what are described in the schematic interpretation. This means that the processing is not effective. This observation has been identified since the first submission of the article and has not been resolved.
  4. Figure 5: An image with raw data is displayed. I fail to see the benefit of showing raw data when processed data suggestive of buried structures can be shown.
  5. Figure 6: (b) part still has gain correction issues. Hyperbolic reflectors can be visible increasing gain. The matGPR plugin has a tool for this. At this moment, as it is, the hyperbolic reflectors are not well observed. It can be improved.
  6. Figure 8: My apologies but I still manage to understand the improvement introduced by processing operations.
    Possibly it is due to the image being a zoom of a larger profile. If it is not possible to improve further, it can be suppressed. Figure 9 shows that there was an improvement introduced by processing.

Author Response

This manuscript is a resubmission of an earlier submission. The following is a list of the peer review reports and author responses from that submission.

Round 1

Reviewer 1 Report

I have significant comments on the peer-reviewed article:

  1. The article describes the basic and well-known principles of the GPR and does not contain new provisions.
  2. The experimental part describes the repeatedly described use of GPR for surveying urban communications, which is also nothing new and interesting.
  3. A list of references in 70 positions is more typical for a review than for a scientific article.

Reviewer 2 Report

Review of the manuscript remotesensing-1289512 “Electromagnetic technique to providing high-resolution RF subsurface images based on Ground Penetrating Radar tool” by Nicoleta Iftimie, Adriana Savin, Rozina Steigmann and Gabriel Silviu Dobrescu.

This is a promising paper about the study of the subsurface, with ground-penetrating radar method, where are located pipes. The used dataset has the common problems, such as low contrast between the ground and the possible buried structures. This implies a better processing workflow to improve the data to allow the interpretation of the buried structures.

The introduction of the manuscript is well designed.  I just suggest that you avoid writing many references as in lines 56, 66, 159 and 231. The cited articles discuss several issues. It is best to focus on the most important references about the issues discussed.

In the line 39, the laser scanning method mentioned must be separated of the ultrasound elastic waves method. The methods are different, the first is geospatial and the second is geophysical. This sentence must be improved.

Another thing I noticed is some repetition of ideas throughout the paper, which should be avoided.

In the Figure 1 (bottom) there is a mistake with the label “reflexion”; must be “reflection”.

The theoretical aspects between the lines 135 and 174 seem to be too detailed. This subject has already been discussed in the respective bibliographical references. Can be abbreviated.

In the line 243 the word “parabola” seems to be a mistake, may be the correct word is “hyperbola”; please verify.

In the line 249, reference [31] don’t use the software GPRMax2D; uses the matGPR software to produce GPR synthetic datasets through FTDT simulations. You can improve the sentence.

In the lines 251, 274 and 322 is mentioned that are used MATLAB code to process the GPR data. I think is better to refer which functions are used. Is not possible to replicate all the processing workflow that you suggest only reading this paper.

Regarding the results (section 3.1) there is a problem. All the processed GPR profiles have the same problem, that is the direct wave. This must be erased from the data in the first processing step. Only with this step is possible to see any contrast change between the ground and buried structures. The results automatically will stay better.

All the section 3.2 seems to be too detailed. This theoretical part also seems to be out of place. It may be mentioned in the method part. However, I once again draw attention to what may be summarized.

The results of the Figure 8 don’t show the improve of the processing operations. The direct wave prevents the good assessment of the processed data.

The Figure 9 can be improved to show the same image before the processing, to compare and verify the effectiveness of the processing operations applied.

The sentence of the lines 361 to 366 have some problem. The GPRMax2D already is a simulation software for GPR data. You can reformulate the sentence to improve the message you can transmit.

Regarding the structure of the paper, I notice a problem. There is not a Discussion section individualized. This must be improved.

Regarding the innovation of the paper, I have some doubts. I can’t identify the proposed innovation. All the described methods are not new. The processing operations mentioned come from bibliographic sources and are well discussed on it. Therefore, I think that the paper should be reformulated, including the title, so that the application of the method to the case study can be presented, showing what is discovered with the performed GPR prospection.

Reviewer 3 Report

Based on the ground penetrating radar evaluation method, this article detects two urban drainage pipes and provides their positions and depths. These positions and depths can be easily found from the reconstructed profile of simulated and actual ground penetrating radar signals.

The techniques and algorithms used in this article allow us to detect the location and depth of underground pipelines with good accuracy. It looks very interesting, but there are still the following shortcomings

  1. The annotation of picture 8 is a bit inaccurate
  2. For picture 6 it should be explained in more detail
  3. Is the pipeline used in this article representative?